# An Enterprise Time Series Forecasting System for Cloud Applications Using Transfer Learning [note 1]

**DOI:** 10.3390/s21051590

**Published:** 2021-02-25

**Authors:** Arnak Poghosyan, Ashot Harutyunyan, Naira Grigoryan, Clement Pang, George Oganesyan, Sirak Ghazaryan, Narek Hovhannisyan

**Affiliations:** 1VMware, Inc., Palo Alto, CA 94304, USA; ngrigoryan@vmware.com (N.G.); pangc@vmware.com (C.P.); goganesyan@vmware.com (G.O.); sghazaryan@vmware.com (S.G.); 2Institute of Mathematics of NAS RA, Yerevan 0019, Armenia; 3TeamViewer Armenia, Yerevan 0018, Armenia; narek.hovhannisyan@teamviewer.com

**Keywords:** time series analysis, anomaly detection, neural networks, hypothesis testing, trend analysis, periodicity analysis, cloud applications, pretrained models, transfer learning, 37M10, 62M45, 62M20, 68T07

## Abstract

The main purpose of an application performance monitoring/management (APM) software is to ensure the highest availability, efficiency and security of applications. An APM software accomplishes the main goals through automation, measurements, analysis and diagnostics. Gartner specifies the three crucial capabilities of APM softwares. The first is an end-user experience monitoring for revealing the interactions of users with application and infrastructure components. The second is application discovery, diagnostics and tracing. The third key component is machine learning (ML) and artificial intelligence (AI) powered data analytics for predictions, anomaly detection, event correlations and root cause analysis. Time series metrics, logs and traces are the three pillars of observability and the valuable source of information for IT operations. Accurate, scalable and robust time series forecasting and anomaly detection are the requested capabilities of the analytics. Approaches based on neural networks (NN) and deep learning gain an increasing popularity due to their flexibility and ability to tackle complex nonlinear problems. However, some of the disadvantages of NN-based models for distributed cloud applications mitigate expectations and require specific approaches. We demonstrate how NN-models, pretrained on a global time series database, can be applied to customer specific data using transfer learning. In general, NN-models adequately operate only on stationary time series. Application to nonstationary time series requires multilayer data processing including hypothesis testing for data categorization, category specific transformations into stationary data, forecasting and backward transformations. We present the mathematical background of this approach and discuss experimental results based on implementation for Wavefront by VMware (an APM software) while monitoring real customer cloud environments.

## 1. Introduction

One of the main goals of IT infrastructure and application monitoring/management solutions is the full visibility into performance, health and security with growing intelligence. The prediction of performance degradations, root cause analysis as well as self-remediation of issues before they affect a customer environment are anticipated features of modern cloud management solutions. Self-driving data centers require the availability of proactive Analytics with AI for IT operations (AIOps) [1] in view of nowadays very large and distributed cloud environments. The key capabilities of the AIOps are predictions, anomaly detection, correlations and root cause analysis on all acquired data including traces, logs and time series (see [2,3,4,5,6,7,8,9,10] with references therein).

Time series collection and analysis is of great importance for various reasons like anomaly detection, anomaly prediction, correlations and capacity planning [11,12,13,14,15]. The administrators of cloud environments require automated forecasts of the future metric-data values for the prediction of the future states of application or infrastructure components. The capacity planning requires trend analysis for resource consumption to predict additional needed processor bandwidth or mass-storage capacity in order to prevent delays and failures. Time series data can also be used for correlation analysis and as the source of anomaly events for further root cause analysis.

Time-series analysis is a significant branch of mathematics and computing that includes a variety of different types of analytical procedures, computational tools and forecasting methods. It is sufficient to mention the well-known and powerful approaches like Fourier analysis, time series decompositions, forecasting by SARIMA and Holt-Winters’ methods (see [16,17,18,19,20] with references therein). However, distributed cloud infrastructures and applications require relatively quick forecasts and are associated with significant temporal constraints, forestalling lengthy and computationally intensive analyses.

In this paper (see also [2]), we focus our attention to time series forecasting with further application to an anomaly detection problem. The application of NN-models and other ML techniques may produce efficient methods [21,22], but a naive implementation in a cloud-computing environment fails to provide adequate response times and would likely be far too expensive for most clients. The training and storing of neural networks are both time-consuming and expensive with respect to the necessary resources (central processor unit (CPU), graphical processor unit (GPU) and memory). Hence, it is not feasible to train those models in demand for the specified time series data. Moreover, it would not be attainable to train and store special-purpose neural networks for all of the different possible types of time series. From the other side, a naive attempt to train a single neural network to analyze all of the various different types of time-series data would also likely fail, since different types of time-series data exhibit different types of behaviors and temporal patterns. A single neural network would need a vast number of nodes and even vaster sets of training data to produce reasonable forecasts for global time-series data. The truth should be somewhere in the middle.

The purpose of the paper is application of NN-based models to time series forecasting in cloud applications. The main idea is in training of a generic NN-model and transferring the acquired knowledge to a customer specific time series data never seen before. This should be the only way of overcoming the challenges regarding the resource utilization (GPU trainings of the networks) as the application of the pretrained model does not require on-demand network training. This solution is feasible if the problem can be narrowed down to some classes of time series data with specific behaviors for which the application of pretrained models are attainable. Moreover, those classes should be large enough to cover the sufficient portion of unseen customer data and specific enough by the behavior to deal with moderate network configurations. We found that the class of stationary time series can be properly handled by NN-models. Unfortunately, this class is not common in the discussed domain. Conversely, the majority of time series data contain nonstationary patterns like trend, seasonality or stochastic behavior. However, the class of stationary time series data can be extended to time series categories which can be transformed into the needed class by some simple transformations. This observation outlines the main idea of our approach through the utilization of a pretrained NN-model with preliminary time series classification and transformation into a stationary data via class-specific rules. We develop the theoretical foundation of the approach and show the results of its realization in a real cloud-computing environment. Implementation and testing are performed in Wavefront by VMware [23]. Wavefront offers a real-time metrics monitoring and analytics platform designed for optimization of cloud and modern applications.

It is worth noting that our main goal is the performance of the approach for cloud environments rather than the accuracy of predictions compared to the well-known classical techniques that perform individual training for each specified time series data in the GPU accelerated environments. For us, the performance is trade-off between accuracy and resource utilization. We observed that the accuracy is comparable to the classical ARMA related approaches while preserving resource consumption on acceptable levels. In particular, the application of the pretrained network to a specified time series in a cloud environment can be performed without GPU acceleration and with moderate number of CPU cores.

One of the important applications of time-series forecasts is the detection or prediction of infrastructure and application performance degradations or failures. Accurate and fast anomaly/outlier detection leads to proactive problem resolution and remediation before it affects a customer environment. This means that timeliness and preciseness of anomaly detection are of great importance for distributed systems. However, it is worth noting that forecasts based anomaly detection may be associated with low response times especially for longer forecast horizons. Moreover, precautionary procedures should be taken for reduction of false positive anomalies that can unnecessary disturb users with alarms.

Another important aspect tightly related to the problem is association of time series outliers with system anomalies which is roughly speaking not always true. In any case, such problems are unsolvable without intrusion of domain expertise into mathematical models or their outcomes. Our solution to anomaly detection utilizes a test window which is smaller than the forecast window for providing adequate response times and meanwhile contains enough data points for the reduction of false positives. The fraction of violations of the confidence bounds of the forecasts in the test window generates an anomaly signal. The anomaly monitor generates alarms and warnings or launch preventive procedures whenever the anomaly score rises above a particular threshold.

The current paper is organized as follows. Section 2 performs literature review for time series forecasting methods, data categorization via hypothesis testing and time series anomaly detection. Section 3 presents the main idea of time series data forecasting via pretrained NN-models for cloud environments. The main barrier towards effective application of NN-models is data stationarity. Section 4 describes hypothesis testing approaches for data categorization. Each category identifies the set of rules that transforms a time series into a stationary one. Section 5 shows the method of application of NN-models to different data categories with corresponding forecasts and confidence bounds. Those bounds serve as baseline for a time series normality behavior, and Section 6 describes its application for time series anomaly/outlier detection. Section 7 outlines the configuration of the pretrained NN-model and the process of its training for a global database of time series data. Section 8 demonstrates the results of implementation of the forecasting engine and performs comparisons with some of the classical approaches. Section 9 provides some conclusive remarks.

## 2. Related Work

Application of pretrained NN-models to solution of different problems is a well-founded approach for many domains like classification, image processing, voice recognition, text mining, etc. (see [24,25,26,27,28] with references therein). It is known as transfer learning for some applications [28] and is a natural approach for knowledge generalization and complexity reduction. Such pretrained networks (VGG, ResNet, BERT, etc.) have deep learning sophisticated architectures requiring serious resources and datasets for their trainings. Application of this idea to time series forecasting is a novel approach. We only consider the first steps and many questions still need clarifications. We trained the simplest networks like MLP or LSTM, but the exact required architecture remains unknown and extended research will be carried out elsewhere.

Time series forecasting is an important area for many diverse areas such as econometrics, signal processing, astrophysics, etc. The classical theory of forecasting [16,17,18] deals with time series data with the wide range of properties. ARMA models are very powerful for stationary time series data. However, in many problems (e.g., economy, business) time series exhibit nonstationary variations due to trend or seasonality (deterministic or stochastic). Models that analyze nonstationary data require knowledge of those patterns. Some models assume that variations are deterministic and apply regression analysis to handle both trend and seasonality. One of the interesting approaches is time series decompositions known as STL [20]. Other approaches model data as having stochastic trend as in ARIMA and stochastic seasonality as in SARIMA. Holt-Winters’ seasonal and SARIMA models represent a broad and flexible class relevant for many applications. It has been found empirically that many time series can be adequately fit by those models, usually with a small number of parameters.

Naturally, models based on artificial neural networks should have better performance due to their nonlinearity, flexibility and ability of generalization [21,29,30]. It was assumed that no any specific assumptions need to be made about the model which should be one of the most important advantages. Different authors showed in their studies and experiments [31] that better results compared to SARIMA and related models could be achieved only by combination of transformations that “stabilize” the behavior (e.g., detrending, deseasoning) of the specified time series [32]. However, the results regarding the forecasting of nonstationary time series data via NN models are very controversial [31,33,34,35].

Time series data categorization is the crucial milestone for our approach. Application of NN-models and deep learning for time series classification should be explored elsewhere [36]. In the current research, we restrict ourselves by the classical hypothesis testing methods for trend and seasonality detection. Each data category identifies the set of transformations that will convert any class-representative into a stationary time series. For example, we detect deterministic and stochastic trends via KPSS-test [37] and ADF-test [38,39,40,41,42], respectively. They have the best combined power for data with moderate length. Deterministic seasonality can be tested via Fourier analysis which treats data in the frequency domain [19]. It reveals only sinusoidal patterns in data. We implemented the simplest approach known as phase dispersion minimization (PDM) test [43,44,45] which treats data in the time domain. It is applicable for data with few observations, with nonregular sampling, with gaps and with nonsinusoidal periodicities. In general, multiseasonality can be tested with several period-lags which should be considered elsewhere. Stochastic seasonality can be tested via CH-test [46], HEGY-test [47], OCSB-test [48] and more [49,50,51,52]. In this article, we only test the deterministic seasonality. Stochastic seasonality will be considered elsewhere.

Time series anomaly/outlier detection has been investigated by numerous authors for many applications [11,12,13,14,53,54,55,56,57,58]. It is known as a very hard problem with many diverse ramifications. NN-based methods and deep learning are becoming very popular and powerful [55,59].

## 3. Main Idea

Application of NN-based models to time series forecasting in cloud applications faces several challenges. One of the main ones is the restriction on the computing resource utilization. Complex network trainings require powerful GPUs and sufficient volume of data. Those are real issues in cloud environments, and the solution is in transfer learning, or in other words, in the utilization of pretrained NN-models. We train a network on a global dataset collected across different customers and store it for further application to a specific time series data definitely never seen before. This means application of the NN-models on-demand for a specified time series via several CPU cores without GPU acceleration. The training of the models will be performed in private powerful data centers with enough GPU resources. Figure 1 shows (see also Figure 1 in [2]) the outline of this idea.

The entire system consists of two separate and independent subsystems called as offline and online engines. The offline engine performs a model training for a network with some predefined configuration and on a global training database containing time series with diverse behaviors. The final weights, together with the configuration, define the pretrained NN-model. We store the configuration and weights in a cloud (as a file in the “json” format) for on-demand access. The global database should be regularly updated by new time series data across different customers and environments. As a result, the pretrained model would be regularly updated. The online engine corresponds to a customer cloud-computing environment. An APM software restores the weights and configuration of the pretrained network from the file. First, the engine retrieves the configuration for data processing. Second, it restores also the weights and applies the model to the processed time series data for a forecasting. The offline mode requires GPU-powered data centers. The online mode is the customer common computing space without a GPU-acceleration.

The diversity of time series data behaviors is a crucial milestone connected with the system of Figure 1 that probably will not allow a naive realization of the approach. We already mentioned the role of data categorization for proper model construction. One of the possible scenarios is the selection of some data classes and the corresponding class-specific network models. Those pretrained networks can be stored and called on-demand. Preliminary data categorization should be performed in both online and offline modes for treating with the required models. This scenario should be considered elsewhere.

Another scenario, developed in this paper, is the selection of a single class that can be adequately treated by a unique NN-model. The other time series data can be treated properly after some preliminary transformations to the specified class. This scenario should be more optimal as only one model should be trained, stored and applied. How does one find the class with the best trainable and transformable characteristics? Our previous discussion indicated that the class of stationary time series should be the first candidate for experiments. They can be properly modeled by NN-models, and the techniques of transformation of a nonstationary time series into a stationary (called before as stabilization) are theoretically well-founded. The set of stabilizing transformations is the class-specific. A deterministic trend can be stabilized by linear or nonlinear regression. A stochastic trend can be removed by a differencing. A seasonality can be removed via seasonal-means or lag-differencing. We apply different well-known hypothesis testing algorithms for time series classification. We use KPSS-test and ADF-test for the detection of deterministic and stochastic trends. We test a deterministic periodicity via PDM-test. A stochastic-periodicity can be verified via CH-test.

As a result, we perform model trainings only for stationary time series data. We have two possibilities. Either collect only stationary time series for a global database or collect all available time series and use them after preliminary stabilization. We implemented the second scenario and the flowchart of Figure 2 describes it. The training dataset contains time series of any behavior. Hypothesis testing engine performs data categorization which defines the set of required transformations. The resulting stationary time series is used for the model training. It is possible that hypothesis testing fails to categorize data. We assign it to an unknown class without further utilization for the training or forecasting.

Application of the pretrained neural network to a user specified time series data will similarly pass through the hypothesis testing engine for data categorization (see Figure 3). The model will be straightforwardly applied to a stationary data. A nonstationary time series should be transformed into a stationary one, and then the corresponding inverse transformations should be applied to the forecast for recovering the original scale and behavior.

The next challenge connected with the application of NN-models is in the limited number of predefined input/output nodes of the networks. It means utilization of a small number of history and forecast data points for training and prediction. In the current implementation, we use networks with 40 inputs and 20 outputs resulting in utilization of 40 history points to get 20 forecast values. It restricts the model capability to use bigger number of history points even when they are available. NN-models require uniformly sampled input points. The forecast points will appear with the same monitoring interval as the history points. Figure 4 shows a sparse grid (the red dots) selected from a history window (red and black dots together) by a comb-like procedure. Our idea is in mobilization of the entire history window as the collection of sparse grids (see Figure 4). We take *N* uniformly sampled history points multiple to the size of the input of the pretrained-network. In our specific implementation, it should be multiple to 40 and N=k∗40 where k=1,2,⋯ can be selected from the complexity considerations. This is the entire history window (see the “full grid” in Figure 4). We divide the full grid into *k* different uniformly sampled sparse grids containing 40 data points by the same comb-like procedure. All sparse grids have the same monitoring interval and together they combine the full grid. The sparse grids will be used independently for a network training and predictions. The corresponding forecast window will contain 20∗k uniformly sampled data points.

It is worth noting that the hypothesis testing should be applied to the entire full grid to have sufficient statistics for revealing the behavior of a time series. Then, the same class-specific transformations should be applied to each sparse grid.

## 4. Hypothesis Testing for Data Categorization

In this paper, we restrict ourselves by some specific data categories that contain deterministic and stochastic trends, deterministic and stochastic-periodicities. In all those cases, we are aware how to transform a nonstationary data into a stationary one with further application of NN-models. The list of categories can be enlarged if the corresponding transformations are known. It should be reasonable to add more domain specific data categories based on some expertise. The flowchart of Figure 5 illustrates the workflow of a time series categorization engine.

The engine starts with the periodicity analysis. It tests for the three data categories called as stationary-periodic, trend-periodic and stochastic-periodic time series. The PDM-test inspects the first two categories and CH-test the last one. The PDM-test starts with the stationary-periodic class. It runs across different lags for a time series, measures their significance (see below) and either assigns the original data to the stationary-periodic class with a specific period-lag or rejects it. In the last case, it tests for the trend-periodic class. The test removes a possible deterministic trend (without checking its existence) via linear regression and verifies a periodicity once more. It runs across different lags for the detrended data and either assigns it to the stationary-periodic class with a specific period-lag or rejects it. In the first case, the original data should be assigned to the trend-periodic class with the known trend component and periodicity-lag. In the second case, the CH-test inspects data for a stochastic periodicity. Time series is assumed to be nonperiodic if all mentioned periodicity tests fail. A nonperiodic time series data should be scanned for a stationarity or trend. The combination of the KPSS-test and ADF-test will classify data into one of the following data categories: stationary, trend-stationary (the trend is deterministic), stochastic-trendy (the trend is stochastic also known as unit-root process) and an unknown type if all other tests fail to recognize a time series behavior. We do not utilize unknown types.

### 4.1. Periodicity Analysis

We restrict ourselves by stationary-periodic and trend-periodic categories. The stochastic-periodic time series will be considered elsewhere although the idea is identical to the discussed.

Let $y_{t}$, t=1,⋯,T be the observed time series data. We say that $\ell_{0}$ is the period-lag for time series $y_{t}$ if
(1)$$y_{t+k*\ell_{0}}=y_{t},\;\;k=1,2,\cdots$$

In reality, we can only expect approximate equality due to noise and instability in a time series
(2)$$y_{t+k*\ell_{0}}\approx y_{t},\;\;k=1,2,\cdots$$

We perform inspection of a periodicity by the PDM-test [43,44,45]. The idea is very simple and Figure 6 illustrates it. It shows a pure periodic time series data with $\ell_{0}=37$ (see (Equation 1)). We consider two different subsequences uniformly sampled from the data. The sampling rate of the first subsequence coincides with the true period-lag $\ell =\ell_{0}=37$ (see the top chart). It is a constant subsequence with zero variance due to the periodicity of the original time series. The sampling rate of the second subsequence does not match the period-lag $\ell \neq \ell_{0}$ (see the bottom chart). The variance of the second subsequence is close to the variance of the original time series. This observation is the cornerstone of the PDM-test.

More precisely, let $\sigma^{2}$ be the variance of time series $y_{t}$
(3)$$\sigma^{2}=\frac{1}{T-1}\sum_{t=1}^{T}{\left(y_{t}-\bar{y}\right)}^{2},$$
where $\bar{y}$ is the average. Assume *M* distinct samples collected from the time series with the same lag *ℓ* and containing $n_{j}$ data points with variances $s_{j}^{2}\left(\ell \right)$, j=1,…,M. We denote by $s^{2}\left(\ell \right)$ the average variance of the samples as follows
(4)$$s^{2}\left(\ell \right)=\frac{\sum_{j=1}^{M}\left(n_{j}-1\right)s_{j}^{2}\left(\ell \right)}{\sum_{j=1}^{M}\left(n_{j}-M\right)}.$$

A preliminary goal is the minimization of s(ℓ) via lag selection.

Let us reformulate the problem that allows more efficient implementation. We define the phase of each data point $y_{i}$ at the time stamp $t_{i}$ by the following expression
(5)$$\Phi_{i}=\frac{t_{i}}{\ell }-\left[\frac{t_{i}}{\ell }\right],\;\;\Phi_{i}\in \left[0,1\right],$$
where [·] stands for the integer part. If data points are sampled regularly, then $t_{i}=i$, i=1,2,⋯. In order to detect data points with similar phases, we divide the full phase interval (0,1) into fixed bins (20 in our experiments) and pick *M* samples from the same bin.

Now, consider the following statistic
(6)$$\theta \left(\ell \right)=\frac{s^{2}\left(\ell \right)}{\sigma^{2}}.$$

If $\ell \neq \ell_{0}$
(7)s(ℓ)≈σandθ(ℓ)≈1.

Otherwise, if $\ell =\ell_{0}$, statistic θ will reach a local minimum compared with neighboring periods, hopefully near zero
(8)θ(ℓ)≈0.

We define “importance” of each lag as follows
(9)importance(ℓ)=1−θ(ℓ).

Time series is assumed to be periodic if one of the local maximums of the importance(ℓ) is greater than a predefined threshold (say 0.6). Period-lag $\ell_{0}$ can be identified as the solution of the following optimization problem
(10)$$\left\{\begin{array}{c} \ell_{0}=\arg\;\min\;s\left(\ell \right)\\ importance(\ell_{0})>threshold \end{array}\right.$$

The PDM-test has another interpretation connected with time series decompositions. Assume the following additive decomposition of a time series depending on a lag=ℓ (details see in [16])
(11)TimeSeries=SeasonalComponent(ℓ)+ResidualTimeSeries(ℓ).

The strength of the seasonal component corresponding to a lag=ℓ can be measured as the fraction of variances
(12)$$\frac{variance(Seasonal\;\;Component(\ell ))}{variance(Time\;\;Series)}$$
which exactly coincides with the importance defined via (Equation 9).

Figure 7 illustrates the process of identification of a trend-periodic time series data. The top chart shows almost periodic time series data (see (Equation 2)) with added trend shown as a straight line. The middle chart shows the graph of importance(ℓ) for the original time series. All local maximums have importances smaller than the threshold. It rejects the periodicity of the original time series. The bottom chart shows the graph of importance(ℓ) for the detrended time series via linear regression. The first local maximum corresponding to lag (ℓ=19) has the importance above the threshold. The time series can be categorized as from the trend-periodic class with $\ell_{0}=19$ period-lag.

### 4.2. Trend Analysis

We restrict ourselves by stationary, trend-stationary and stochastic-trendy (a unit-root process) categories. Data categorization will be performed via KPSS-test and ADF-test. We follow [37,38,39]. Let $y_{t}$, t=1,⋯,T be the observed time series data. The KPSS-test considers the following decomposition of $y_{t}$ into the sum of a deterministic trend, a random walk and a stationary error
(13)$$y_{t}=\xi t+r_{t}+\varepsilon_{t},$$
where $r_{t}$ is the random walk
(14)$$r_{t}=r_{t-1}+u_{t}$$
with $r_{0}=c$ and $u_{t}$ from $iid(0,\sigma_{u}^{2})$. Under the null-hypothesis, $y_{t}$ is trend-stationary if $\sigma_{u}^{2}=0$. We call this test as $KPSS_{ct}$. In a special case ξ=0, $y_{t}$ will be stationary around a level *c* or simply, a stationary. We call the test as $KPSS_{c}$. Thus, we consider the following two hypothesis testing scenarios:(15)$$\begin{array}{cc} {\mathrm{KPSS}}_{c}\;\mathrm{test}: & \\ -\;Null\;Hypothesis: & \;stationarity\\ -\;Alternative\;Hypothesis: & \;unit-root\;process \end{array}$$
and
(16)$$\begin{array}{cc} {\mathrm{KPSS}}_{\mathrm{ct}}\;\;\mathrm{test}: & \\ -\;Null\;Hypothesis: & \;trend\;stationarity\\ -\;Alternative\;Hypothesis: & \;unit-root\;process \end{array}$$

Both tests apply ordinary least squares (OLS) for coefficient determination and let $e_{t}$, t=1,2...,T be the corresponding residuals. The idea of the KPSS-test is in verification of the hypothesis $\sigma_{u}^{2}=0$ via LM-statistic defined as
(17)$$LM=\sum_{i=1}^{T}\frac{S_{i}^{2}}{{\hat{\sigma }}_{\varepsilon }^{2}},$$
where
(18)$$S_{t}=\sum_{i=1}^{t}e_{i},\;\;t=1,2,\;\ldots ,\;T,$$
and ${\hat{\sigma }}_{\varepsilon }^{2}$ is the estimate of variance of $\varepsilon_{t}$ (see (Equation 13)). The corresponding *p*-values ($P_{v}$) can be found in [37].

The ADF-test uses the following data model for time series $y_{t}$
(19)$$\Delta y_{t}=y_{t}-y_{t-1}=c+\alpha_{0}y_{t-1}+\sum_{s=1}^{m}\alpha_{s}\Delta y_{t-s}+\varepsilon_{t},$$
where *c* is the level, $\varepsilon_{t}$ is a stationary process and *m* is the order of the model. The value m=0 corresponds to DF-test (Dickey–Fuller test). ADF-test uses OLS for the coefficients determination. Akaike information criterion is used for the order selection. The test is carried out under the null hypothesis $\alpha_{0}=0$. Alternatively, test verifies the condition $\alpha_{0}<0$ that corresponds to a stationary process. We call this test as $ADF_{c}$:(20)$$\begin{array}{cc} {\mathrm{ADF}}_{c}: & \\ -Null\;Hypothesis: & \;unit-root\;process\\ -Alternative\;Hypothesis: & \;stationarity \end{array}$$

The value c=0 corresponds to a unit-root process without a drift. The $ADF_{c}$-test applies DF-statistic [38] for determination of the corresponding *p*-values ($P_{v}$).

The flowchart of Figure 8 illustrates the workflow for trend and stationarity testing. It shows the priority of test applications. We sequentially apply $KPSS_{c}$, $KPSS_{ct}$, $ADF_{c}$ and inspect the corresponding *p*-values of the tests. If the corresponding $P_{v}\geq 0.05$ (corresponding to 95% confidence level), we stop the procedure and categorize data accordingly. The data type is unknown if all tests fail. Figure 9 shows application of this flow for specific examples.

Figure 9 presents the corresponding *p*-values of the tests. We use in our experiments implementation of $KPSS_{c}$, $KPSS_{ct}$ and $ADF_{c}$ from Python module “statsmodels”. The *p*-value of the $KPSS_{c}$-test for the first example is larger than 0.05. Although, the *p*-value of the $KPSS_{ct}$ is also bigger than 0.05, the priority order categorizes data as the stationary. By the way, in the production, we don’t need to verify the second *p*-value if the first one is already the winner. The *p*-value of the $KPSS_{c}$-test for the second example is smaller than 0.05 and the categorization engine rejects the stationarity (null hypothesis). The *p*-value of $KPPS_{ct}$-test equals to 0.05. We cannot reject the null hypothesis by 95% confidence and classify data as from the trend-stationary class. However, it is interesting to see that the *p*-value of the $ADF_{c}$-test is also very big. This means that data can be categorized either as trend-stationary or as stochastic-trendy, but the priority order (actually based on the simplicity principle) selects the trend-stationarity as the winner. For the third example, the *p*-values of $KPSS_{c}$ and $KPSS_{ct}$ tests are smaller than 0.05. The time series belongs to the stochastic-trendy class as the $P_{v}$ of the $ADF_{c}$-test is bigger than 0.05.

## 5. Time Series Forecasts with Confidence Bounds

In this section, we discuss the application of the pretrained NN-models to time series forecasting in the online mode. The entire engine was described in Figure 3. Recall that NN-models were only pretrained for stationary time series data. A nonstationary time series should be converted into a stationary one and the corresponding forecast should be reconverted by backward transformations. We restrict ourselves by the five specific data types mentioned before: stationary, trend-stationary (deterministic trend), stochastic-trendy (unit-root process), stationary-periodic and trend-periodic. More extended version of data categorization will be considered elsewhere.

### 5.1. Stationary Time Series Data

We treat with the class of stationary time series directly without forward and backward transformations. We only perform scaling into interval [0,1] as the well-known standard procedure before application of neural networks. We apply the following transformation for the scaling
(21)$$X_{scaled}=\frac{X-X_{min}}{X_{max}-X_{min}},$$
where *X* is the original time series and $X_{min}$, $X_{max}$ are its minimum and maximum values, respectively. We apply reverse scaling to the corresponding forecast by multiplying with $\left(X_{max}-X_{min}\right)$ and summing by $X_{min}$.

The confidence bounds of the forecasts are one of the important pieces of any approach. Ideally, we need rather long historical data with different forecasts that will help to extract the bounds based on some confidence levels (say 95%). Unfortunately, we have only one history window with the corresponding forecast window and forced to extrapolate that available information to future data points without a strong evidence.

Assume $y_{k}$, k=1,⋯,T be the observed data points from a stationary time series, $\mu =mean\{y_{k}\}$ be the average and ${\hat{y}}_{j}$, $j=T+1,\;\cdots ,\;T+T_{f}$ be the corresponding forecast values. Let $y_{s}^{high}$, $s=1,\;\cdots ,\;N_{high}$ be the data points from the history window that are bigger or equal than μ. Similarly, let $y_{s}^{low}$, $s=1,\;\cdots ,\;N_{low}$ be the data points from the history window that are smaller or equal than μ. Then, the higher and lower “standard deviations” can be estimated as follows
(22)$$\sigma_{high}={\left(\frac{1}{N_{high}-1}\sum_{s=1}^{N_{high}}{\left(y_{s}^{high}-\mu \right)}^{2}\right)}^{\frac{1}{2}},$$
and
(23)$$\sigma_{low}={\left(\frac{1}{N_{low}-1}\sum_{s=1}^{N_{low}}{\left(y_{s}^{low}-\mu \right)}^{2}\right)}^{\frac{1}{2}},$$
respectively. We define upper and lower bounds (UB and LB as confidence bound vectors) for each of the point in the forecast window as follows
(24)$$UB_{j}={\hat{y}}_{j}+z*\sigma_{high},\;\;LB_{j}={\hat{y}}_{j}-z*\sigma_{low},\;\;j=T+1,\;\cdots ,\;T+T_{f},$$
respectively, where parameter *z* stands for the criticality levels z=1,2,3,⋯.

The confidence bounds for any data category can be found similarly. We perform forward transformations for converting a time series into a stationary, calculate the corresponding forecast with the confidence bounds and apply the backward transformations.

Figure 10 illustrates an example of a stationary time series. The history window contains 400 points (see the green curve in Figure 10). The $KPSS_{c}$-test applied to the history window returns the *p*-value larger than 0.05 and the null hypothesis for data stationarity cannot be rejected. This means application of the pretrained neural network to the sparse grids without any forward/backward transformations. The history consists of 10 sparse grids with 40 points in each. They provide with 10 forecasts with 20 points in each. We collect those forecasts with the confidence bounds together and get 200 points in the forecast window (see the red curve in Figure 10). The gray area corresponds to the confidence bounds with z=3 (see (Equation 24)). We see from the figure that the forecast is smoother (less variable) than the observed data points which is common for the NN-based models. Only a few data points violate the confidence bounds. There are no changes or anomalies/outliers in data to report.

### 5.2. Trend-Stationary Time Series Data

The class contains time series data with deterministic linear trend that can be removed via ordinary least squares (linear regression). In general, a nonlinear trend also can be considered along the same ideas. The trend removal procedure should be applied to each sparse grid with the further calculation of the forecasts and confidence bounds. The original behavior should be recovered via trend addition as the backward transformation applied to each sparse grid.

Figure 11 shows an example of a trend-stationary time series. We apply the $KPSS_{c}$-test to the history window (see the green curve in Figure 11). Its *p*-value is smaller than 0.05 and the null-hypothesis is rejected. This means that the time series is not stationary and the $KPSS_{ct}$-test should be tried. Its *p*-value is larger than 0.05. We cannot reject the null-hypothesis and assume that data is from the trend-stationary class. The linear regression is applied to each sparse grid for identifying the corresponding slopes ($k_{s}$) and intercepts ($b_{s}$). We remove the trends $\left(k_{s}t+b_{s}\right)$ from all sparse grids. There are 7 sparse grids with 40 points in each. We apply the pretrained NN-model to each detrended time series on the sparse grids and get the forecasts with the confidence bounds. Finally, the forecasts, together with the bounds, are modified by the backward transformations as trend additions. As a result, we get 140 forecast data points shown as a red curve in Figure 11. The gray area shows the final confidence bounds. Almost all observed data points lay between the bounds. There are no changes or outliers in data to report which is in line with our visual perception.

### 5.3. Stochastic-Trendy Time Series Data

The class contains time series data that can be transformed into a stationary via differencing. Those time series are known also as unit-root processes or unit-root processes with a drift. The differencing should be applied to each sparse grid with further calculation of the forecasts and confidence bounds. The original behavior should be restored via backward-differencing.

Figure 12 shows an example of a time series with the stochastic trend.

We sequentially apply $KPSS_{c}$ and $KPSS_{ct}$ tests to the entire history window (see the green curve in Figure 12). Both *p*-values are smaller than 0.05 and both null-hypotheses are rejected. The *p*-value of the $ADF_{c}$-test is larger than 0.05. We cannot reject the null-hypothesis and categorize data as from the stochastic-trendy class.

The process of forecasting for the stochastic-trendy class has some peculiarities compared to the previous examples. Our implementation of the pretrained model requires 40 inputs. However, after the differencing, the number of points in a grid will be reduced by one. That is why 41 points should be selected in a sparse grid as the starting point. The history window in Figure 12 (the green curve) contains 7 sparse grids with 41 points in each. Let $v_{t}$, t=0,⋯,40 be the time series across one of the sparse grids. We assume the following model for $v_{t}$
(25)$$v_{t}-v_{t-1}=c+u_{t},\;\;t=1,\;\cdots ,\;40,$$
where *c* is the intercept (responsible for a drift) and $u_{t}$ is a stationary process. Let $z_{t}$ be the differenced time series
(26)$$z_{t}=v_{t}-v_{t-1},\;\;t=1,\;\cdots ,\;40$$
with the corresponding model
(27)$$z_{t}=c+u_{t},\;\;t=1,\;\cdots ,\;40.$$

This means that $z_{t}$ is a stationary time series containing 40 data points for application of the pretrained model. Let ${\hat{z}}_{40+j}$, j=1,⋯,20 be the corresponding forecast. The forecast ${\hat{v}}_{40+j}$, j=1,⋯,20 for the original time series can be derived via application of the backward-differencing
(28)$${\hat{v}}_{41}=v_{40}+{\hat{z}}_{41},$$
and
(29)$${\hat{v}}_{40+j}={\hat{v}}_{40+j-1}+{\hat{z}}_{40+j},\;j=2,\;\cdots ,\;20.$$

The same procedure should be applied to confidence bounds and other sparse grids. Those forecasts and bounds are shown in Figure 12 as red curve and gray area, respectively. Normally, the forecast window for time series from this class should be rather short. We see that data points closer to the current time lay within the bounds while the farther points mostly lay outside. It is a normal behavior for time series data with the stochastic trend due to its unpredictable behavior.

### 5.4. Stationary-Periodic Time Series Data

The class contains periodic (see (Equation 1)) or almost periodic (see (Equation 2)) time series with known period-lags $\ell_{0}$. We consider two different approaches for the corresponding forecasts.

The first approach is connected with the structure of history windows described above. The number of points in a history window is multiple to the size of the input of the pretrained NN-model. We already mentioned that the current model has 40 inputs and the history window will have 40∗k data points with any k=1,2,⋯. Hence, we have *k* different sparse grids for separate forecasts. The selection of *k* should be a trade-off between the needed resolution and the complexity of computations. The first approach requires selection $k=\ell_{0}$. A stationary-periodic time series with period-lag $\ell_{0}$ will be almost a constant across all sparse grids sampled with the same lag (see Figure 6) allowing us direct application of the pretrained model.

Figure 13 shows an example of a stationary-periodic time series with $\ell_{0}=19$ revealed via PDM-test, where the forecasts are estimated based on the mentioned procedure. The history window contains 40∗19=760 data points (we show only the part of it). The forecast window contains 380 data points. We also show the data points of the history window for better visual perception.

The mentioned approach may cause problems due to several reasons. The first problem is the connection of the history and forecast windows with the period-lag. The latest can be rather large leading to unreasonable extensive computations. The second problem is in unknown period-lag. It means time series sampling with some preselected *k* and then resampling according to its period. It will cause time consuming duplicated data processing.

The second approach does not require the connection between the number of sparse grids and the period-lag. A time series can be sampled with any value of parameter *k*. Deseasoning of the time series can be performed via seasonal means (details see in [16]). We can remove the seasonal component, apply the pretrained neural network and return the seasonal component back. Similarly, the confidence bounds can be estimated. The forecast will be identical to the one presented in Figure 13.

### 5.5. Trend-Periodic Time Series Data

This class contains periodic or almost periodic time series with some linear trend. The process of data categorization was discussed before. We need to remove the detected trend, get the forecast as it was discussed in the previous subsection and return the trend back. Figure 14 shows an example of a time series from the trend-periodic class. We applied the PDM-test to the history window (see the green curve) before and after the linear trend removal. In the first case, the PDM-test fails to detect any significant period-lag. In the second case, it observed a periodicity with period-lag $\ell_{0}=19$.

We take 19 sparse grids with 40 data points in each and directly apply the pretrained NN-model, according to the first approach of the previous subsection, to the detrended time series. Later, we recover the removed trend for the forecast and confidence bounds. We see that almost all observed data points are within the confidence bounds.

## 6. Anomaly Signals from Time Series Data Forecasts

In this section, we discuss an approach to anomaly signal time series generation based on the confidence bounds of the forecasts. Each data point in the anomaly signal shows the percentage (fraction) of observed data points in a test window that violate upper and/or lower confidence bounds. The anomaly signal may detect or predict anomalous conditions whenever its values exceed a particular threshold. In such situations, an anomaly monitor will generate alarms indicating some behavioral changes in a specified time series. We consider details for NN-based forecasting methods described in the previous sections, although the approach is applicable to any predictive model.

One of the principle problems in time series data anomaly/outlier detection is setting of the proper trade-off between the timeliness and confidence of the detections. From the one side, the alarms should be detected as faster as possible for preventive actions before the alarms will impact customers’ environments. From the other side, the big number of false positive alarms overwhelms system administrators and decreases the confidence towards the anomaly detection system. The trade-off may be resolved by the proper selection of underlying parameters for the anomaly signal generation.

An initial indication that the state of a monitored system has begun to change in an unexpected fashion is that an observed data point diverges from its forecast. However, no one is expecting that this single indication will be used as a detectable signal due to a significant noise in time series data and its nondeterministic nature which makes very accurate predictions impossible. Another indication can be violation of a confidence bound by an observed data point. Nevertheless, no one will pay attention to that signal if the subsequent observed time series data are within the bounds or even close to the predicted values. The violation may possibly be an outlier due to noise or some sudden instability rather than an indication of a serious malfunctioning of a system. It is likely that many false-positive alarms will appear if alarms and warnings will be generated based on single-data-point or short-term departures of observed time series data values from the forecast ones. However, by waiting until a pattern of detected preliminary behavioral change will emerge, the problem may have already cascaded to a point when proactive actions can no longer be possible due to some catastrophic impacts on the system. The period of time between the initial indication of an anomaly and the onset of serious degradation depends on the nature of time series and the process that it describes.

Figure 15 and Figure 16 illustrate the basis of the solution to the mentioned problems. Recall that there are three forecast time series data that should be used for anomaly signal generation. The first one is the predicted time series (forecast window) and the next two are predicted upper and lower bounds. We are not showing the last two time series data in the figures for the simplicity but the term violation always refers to the bounds. In addition, we refer to a history window which contains time series data points from which the mentioned forecasts were calculated. Moreover, observation window contains actually monitored time series data points. It is assumed that the history and forecast windows contain uniformly sampled time series data points with the same monitoring intervals.

Figure 15 represents the hidden background of an anomaly monitor while following a specific time series data. The monitoring will be started by indication of the length of the forecast window. We describe below the process of parameter selection in more details. Now, we assume that the forecast window contains *T* uniformly sampled data points. To be more precise, parameter *T* must be a multiple of the size of the output of the pretrained NN-model. Moreover, in the previous sections it was indicated the strict connection between the sizes of a history window and the corresponding forecast window.

The current pretrained network uses 40 historical points to predict 20 future points. It means that the history window is twice as long as the forecast window for the current model. For generality, assume that a history window is *r* times longer than a forecast window (see Figure 15). The user will not see the history window. His UI chart will contain several forecast windows, as much as possible to fit. We show *m* such intervals in Figure 15. The engine will calculate the forecast for the first window and the corresponding anomaly signal will be estimated for all observed data points. Then, the engine will repeat the process for the other forecast windows by shifting the history window to the right by *T* data points until it will reach the final forecast window. There are different reasons why we did not calculate a unique forecast for the entire UI chart (for m·T data points). The first reason is the complexity of data preprocessing. If a user opens a rather big UI chart (*T* is big), then the forecast engine will fail to process r·T data points. The second reason is the desire of immediate incorporation of the latest observed data points into the process of anomaly signal generation.

Figure 16 shows the process of calculation of the anomaly signal for each of the forecast windows. Moreover, the anomaly score must be assigned to each data point in a forecast window. As the entire forecast window can be rather large and by recalling the requirement for the timeliness, we incorporate a test window (smaller or equal to the forecast window) for the percentage calculation (see the “blue” rectangles in Figure 16) for faster reaction to possible anomalies. To each just observed time series data point a test window should be assigned extending to the left by the time axis where the point of interest is the last point of the window. The percentage of violations in the test window is the anomaly score of that last point. Then, the anomaly monitor can visualize the anomaly signal or trigger an alarm based on some threshold value (say 0.8).

Figure 17, Figure 18 and Figure 19 show some specific time series data with the corresponding anomaly signals. “Blue” curves correspond to time series data and “red” ones to the anomaly signals. The values of time series data are shown on the left y-axes, and the values of the anomaly scores on the right. Anomaly scores take values from interval [0,1]. Value 0 means that all observed data points in the test window arrived within the confidence bounds. Value 1 means that all observed data points in the test window violated the confidence bounds.

Let us explain some peculiarities regarding time series visualization in Wavefront. Figure 17, Figure 18 and Figure 19 refer to the Wavefront UI. The UI cannot handle all time series data points available in a database and it applies a method known as summarizing. The figures show that in the current situation the summarizing function uses averaging of data points within a bucket with 7200 s duration. However, NN-model utilizes totally different data points derived from the database via interpolation for uniform sampling. Unfortunately, it means that the actual time series data utilized by the NN-model is not the one that we see in the UI. This was one of the big challenges for the current implementation as the situation should be explained to our product users for increasing the confidence towards the forecasts and anomaly detection visualization.

Figure 17 shows the example of a stationary time series data without visible outliers/anomalies and the corresponding anomaly score is almost flat near the zero value. Small fluctuations in the anomaly score are outcomes of random outliers that go out of confidence bounds but are not visible due to Wavefront data summarization procedure. Figure 18 shows piecewise constant data with two change points. In both cases the anomaly score detect the behavioral changes with the values bigger than 0.8 (the threshold for an alarm generation). It is important that the jumps in the anomaly scores ideally coincide with the jumps in time series data. Figure 19 shows almost constant time series data with some spikes. The behavior of the anomaly signal mimics those spikes. In two cases, the scores became bigger than 0.8, so alarms should be announced. In other cases, the changes and spikes should be ignored.

The biggest problem that the Wavefront customers encountered while consuming the described system for anomaly detection was the large number of false positive alarms. Our experience shows that the customers agree with the reduction of false positives even at the expense of the rising number of false negatives. The common approach to reduction of false positives is through smoothing methods. Paper [60] describes such a kernel-smoothing simple procedure. The kernel smoothing can be applied both to time series data and/or anomaly scores. It actually performs a weighted averaging of data points or anomaly scores where the weights can be extracted via some kernel function. The Gaussian kernel is the most common kernel
(30)$$K_{h}\left(x,y\right)=exp\left(-\alpha \frac{||x-{y||}^{2}}{h}\right),$$
where *h* is the width (window) of the kernel, α>0 is some sensitivity parameter and ∥·∥ stands for the Euclidian distance. Assume that $x_{i}$ are time series data points and $s_{i}$ are the corresponding anomaly scores. A new anomaly score ${\hat{s}}_{i}$ ([60]) is estimated as follows (the weighted mean of anomaly scores detected before the current time)
$${\hat{s}}_{i}=\frac{\sum_{j=i-n}^{i}K_{h}\left(x_{j},x_{i}\right)s_{j}}{\sum_{j=i-n}^{i}K_{h}\left(x_{j},x_{i}\right)},$$
where *n* is the number of points within the window *h*. We can set *h* to be equal to the test window mentioned above. It is possible to calculate two-sided averages if time allows us to wait for new data points to arrive. Similarly, instead of the anomaly scores, we can smooth time series data points. Let ${\hat{x}}_{i}$ be the estimate:$${\hat{x}}_{i}=\frac{\sum_{j=i-n}^{i}K_{h}\left(x_{j},x_{i}\right)x_{j}}{\sum_{j=i-n}^{i}K_{h}\left(x_{j},x_{i}\right)}.$$

Then, a new anomaly score estimate ${\hat{s}}_{i}$ based on ${\hat{x}}_{i}$ can be calculated. Experiments showed that the first approach is preferable. However, more experiments should be performed for the final decision. Probably, time series category (semiconstant, trendy, etc.) should be important for the approach.

## 7. Materials and Methods

In this section, we introduce the NN-model training process in the offline mode. The training was performed in VMware private data centers equipped with powerful GPUs. However, our experimental training database is not big. It includes around 3300 time series, taken from real customer cloud environments. The database contains around 1500-“cpu”, 400-“disk”, 110-“IOps”, 320-“memory”, 450-“network bandwidth”, 100-“network packets” and 410-”workload” metrics. Metrics in the database have 1-min monitoring interval and, in average, 1-month duration. It does not contain any specific information crucial for the model training and similar results should be possible to get via other datasets of time series. Moreover, interesting should be application of synthetic datasets of time series.

The current network has 40 inputs and 20 outputs. We experimented with different dimensions without any serious difference. We noticed that longer input compared to the output resulted in better forecast accuracy. Taking into account the grid structure, we can utilize 40∗k data points in history window to estimate 20∗k points in the forecast window with k=1,2,⋯. We applied a sliding window containing 600 points to each time series. The sliding window had 400=10∗40 history points and 200=10∗20 forecast points. We performed hypothesis testing to the entire sliding window, identified needed transformations and applied those transformations to each sparse grid containing 60 training data points (presumably stationary) for the network input (40 points) and output (20 points).

We tried different network architectures. The first was LSTM networks with stateless configuration and 2 hidden layers with 256 nodes in each. The next was MLP networks with identical configuration. We did’n find significant differences between LSTM and MLP networks for our small dataset. The current model is the MLP network which is very easy to implement without special libraries. We used “relu” activation function for the hidden layers and “linear” activation for the output layer. ‘Adam’ optimizer and mean average error (“mae”) as a loss function were used. We applied 5 epochs for each time series and 20 epochs for the entire database and batchsize=1500. The idea was in getting a generic model for the entire database. The trainings took from several hours to a day depending on the available GPUs.

## 8. Discussion

We tried different implementations of the online mode in Java as enterprise cloud service. The first attempt was utilization of Deep Learning for Java (DL4J) library [61]. It caused some problems due to bigger memory consumption and longer response time. The second attempt was total independent implementation of the MLP network without external libraries. The latest approach is more reasonable as the online mode does not require on-demand trainings and complete deep learning functionality of DL4J is wasteless. Figure 20 shows comparison of timings for both implementations while forecasting a stationary time series. The y-axis shows the timings in milliseconds. The x-axis illustrates different runs for averaging purposes. We see that “DL4J” is far behind compared to “MLP” especially while loading the library.

We performed some comparisons of the current model and classical ARIMA (our internal implementation without the periodicity analysis for both approaches). We applied both models to a database of time series data from our internal cloud environments with different history windows sliding across the time axis. We experimented with 120 points (2 h), 1440 points (1 day), 11,520 points (1 week) and 30,240 points (2 weeks). We calculated the corresponding root mean square relative errors (RMSRE) for each forecast. Table of Figure 21 summarizes the results. It shows overall 279,148 forecast cases. Each column shows the number and percentage of the forecasts for which the corresponding RMSRE is smaller than the mentioned value 0.5,1,2,5 and 10. For example, the last column of the table shows that 126,682 forecasts via NN-model or 45% of all cases have errors smaller than 0.5 while for the ARIMA the same number is bigger by 1%. The difference is insignificant. On average, both methods perform similarly, although ARIMA is slightly better as it was expected.

It should be interesting to compare the average errors across all metrics from the same class. For example, for the class of stationary metrics, NN-model shows an average error 1.2 while ARIMA shows 1.3. For the class of trend-stationary time series, NN-model has an average error 1.57 while ARIMA has 1.52.

Figure 22 represents an example from the Wavefront AI Genie UI [62]. It illustrates the online mode for a specific time series data. AI Genie UI of Wavefront simplifies and automates time series forecasting and anomaly detection capabilities. It requires minimal set of parameters to start running the AI engine. A user can specify (or use defaults) a time series, select the forecast period and the corresponding sensitivity of the confidence bounds. In Figure 22, the red curve corresponds to the historical data, the black curve to the forecast, and the green area to the confidence bounds. The forecast window is 1 week. It means that the history window is 2 weeks as the pretrained network works with 2:1 ratio. Confidence bounds correspond to “moderate” setting (the others are “conservative” and “aggressive”).

The current NN-model uses 4000 data points (uniformly sampled via interpolation) for 2 weeks history, 8640 points for 2 months history and 12,960 points for 6 months. Those selections are the trade-offs between the complexity and grid density. We think that those numbers can be reduced without affecting the accuracy especially for some data categories.

## 9. Conclusions and Future Work

We considered application of NN-based models to time series forecasting and anomaly detection in cloud applications. Throughout the paper, we discussed approaches for overcoming some of the challenges.

The first and main challenge is restrictions on resource consumption in distributed cloud environments. Neural networks require intensive GPU utilization and sufficient data volume which make on-demand training and application of NN-based models unrealistic due to additional costs and unacceptable response times. We proposed a solution along with the ideas of transfer learning. We generate a global database for time series data collected across different cloud environments and customers, train a model in a private GPU-accelerated data centers and apply the acquired knowledge in the form of a pretrained model to a user specified time series data never seen before without GPU utilization. The weights and configuration of the pretrained network are stored in a cloud and monitoring tools can easily access the corresponding files and retrieve the required information for on-demand application to forecasting and anomaly detection.

The second challenge is the weakness of NN-models for analyzing nonstationary time series data. It is a well-known problem and many researchers suggest application of stabilizing procedures like detrending and deseasoning before feeding the network. The stabilizing transformations convert a nonstationary time series into a stationary one, and properly trained NN-models can adequately treat those metrics. We utilize this common idea and train models only for stationary time series. We detect the stabilizing transformations via hypothesis testing. In the offline mode, we perform hypothesis testing to all time series data within the database for finding the set of required transformations for all examples. Those transformations convert all nonstationary time series data into stationary ones before sending to a model for the training. In the online mode, we transform a user specified time series into a stationary data, calculate the corresponding forecast and by application of the backward transformations return to the original scale and behavior. Throughout the paper, we demonstrated the main capabilities of the approach. Moreover, the approach was implemented as a SaaS solution for Wavefront by VMware and it passed full validation in real cloud environments. Our customers mainly utilize the service for anomaly detection.

However, many questions need further investigations. One of the key problems is improvement of the current approach via different networks and configurations. The second interesting problem should be hypothesis testing via NN-based models. We already received some results with one-dimensional convolutional neural networks for data classification. It should be natural to combine both networks to automate data categorization and forecasting. Another interesting problem is designing new models for some new classes of time series data that should improve the accuracy. We also need to check whether bigger datasets will improve the accuracy of the current models.

It is not fair to compare the proposed approach with the methods that train network models in demand for a specific time series data. Undoubtedly, the latest will be more accurate or at least comparable to our approach. Our main goal is the balance between the power and resource utilization. We aimed to develop methods for cloud environments without consumption of valuable resources and with acceptable accuracy.

## 10. Patents

Pang, C. Anomaly detection on time series data. Filed: 22 August 2018. Application No.: 16/109324. Published: 30 January 2020. Publication No.: 2020/0034733A1.

Pang, C. Visualization of anomalies in time series data. Filed: 22 August 2018. Application No.: 16/109364. Published: 30 January 2020. Publication No.: 2020/0035001A1.

Poghosyan, A.V.; Pang, C.; Harutyunyan, A.N.; Grigoryan, N.M. Processes and systems for forecasting metric data and anomaly detection in a distributed computing system. Filed: 17 January 2019. Application No: 16/250831. Published: 27 February 2020. Publication No.: 2020/0065213A1.

Poghosyan, A.V.; Hovhannisyan, N.; Ghazaryan, S.; Oganesyan, G.; Pang, C.; Harutyunyan, A.N.; Grigoryan, N.M. Neural-network-based methods and systems that generate forecasts from time-series data. Filed: 14 January 2020. Application No.: 16/742594.

Poghosyan, A.V.; Harutyunyan, A.N.; Grigoryan, N.M.; Pang, C.; Oganesyan, G.; Ghazaryan, S.; Hovhannisyan, N. Neural-network-based methods and systems that generate anomaly signals from forecasts in time-series data. Filed: 19 December 2020. Application No.: 17/128089. This application is a continuation-in-part of US Application No. 16/742594, filed 14 January 2020.

Poghosyan, A.V.; Harutyunyan, A.N.; Grigoryan, N.M.; Pang, C.; Oganesyan, G.; Ghazaryan, S.; Hovhannisyan, N. Neural-network-based methods and systems that generate anomaly signals from forecasts in time-series data. Filed: 18 January 2021. Application No.: 17/151610. This application is a continuation-in-part of US Application No. 16/742594, filed 14 January 2020.

## Figures and Tables

**Figure 1 sensors-21-01590-f001:**
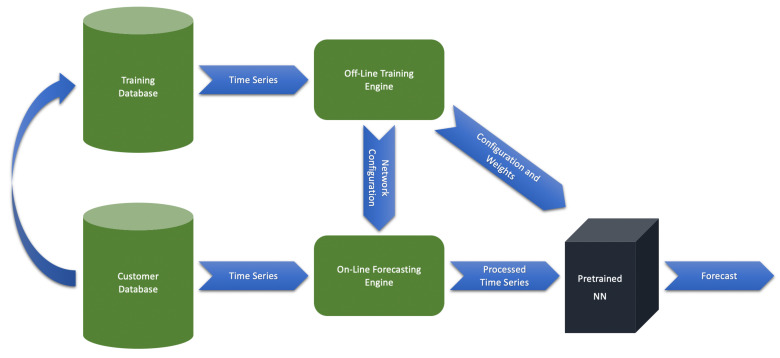
Utilization of pretrained NN-models for cloud applications.

**Figure 2 sensors-21-01590-f002:**
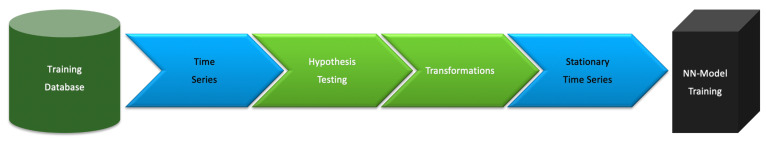
Offline training of an NN-model for stationary time series data.

**Figure 3 sensors-21-01590-f003:**
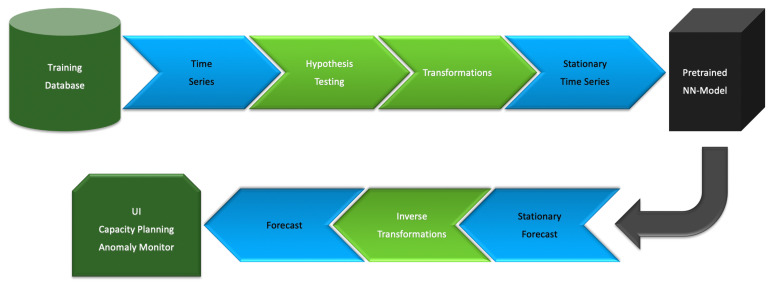
Online forecast for a user specified time series data.

**Figure 4 sensors-21-01590-f004:**
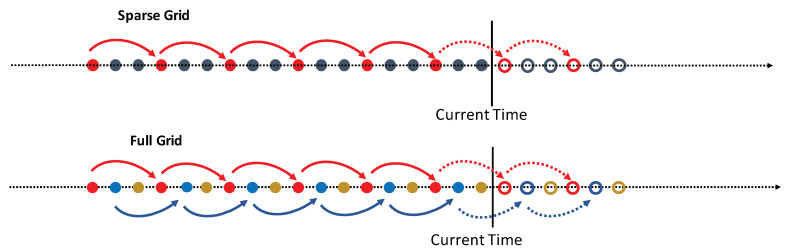
A full grid generation as the collection of sparse grids ready for a network utilization.

**Figure 5 sensors-21-01590-f005:**
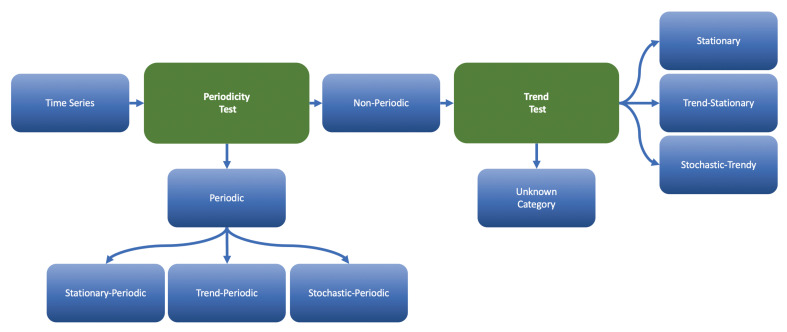
Data categorization engine.

**Figure 6 sensors-21-01590-f006:**
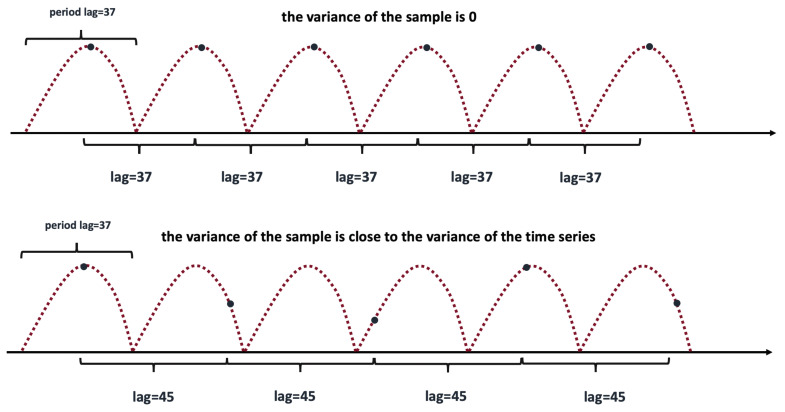
Illustration of the idea of the phase dispersion minimization (PDM) test.

**Figure 7 sensors-21-01590-f007:**
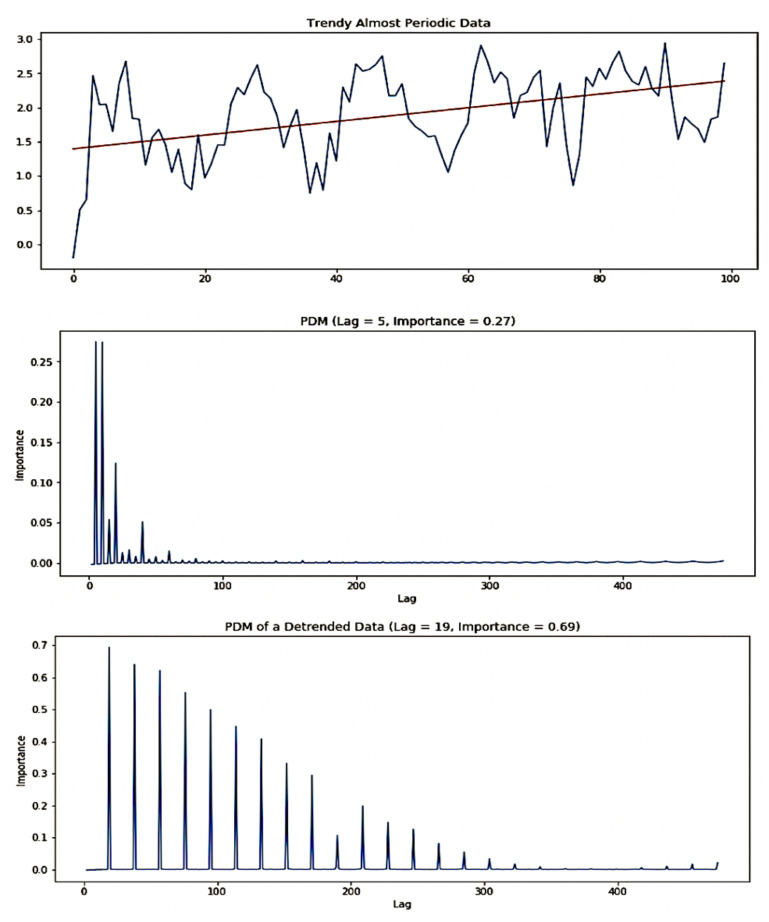
Categorization of a trend-periodic time series.

**Figure 8 sensors-21-01590-f008:**
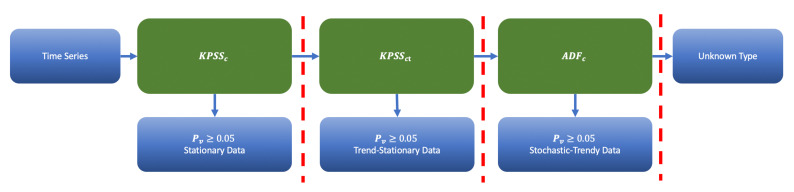
The priority order for a trend detection.

**Figure 9 sensors-21-01590-f009:**
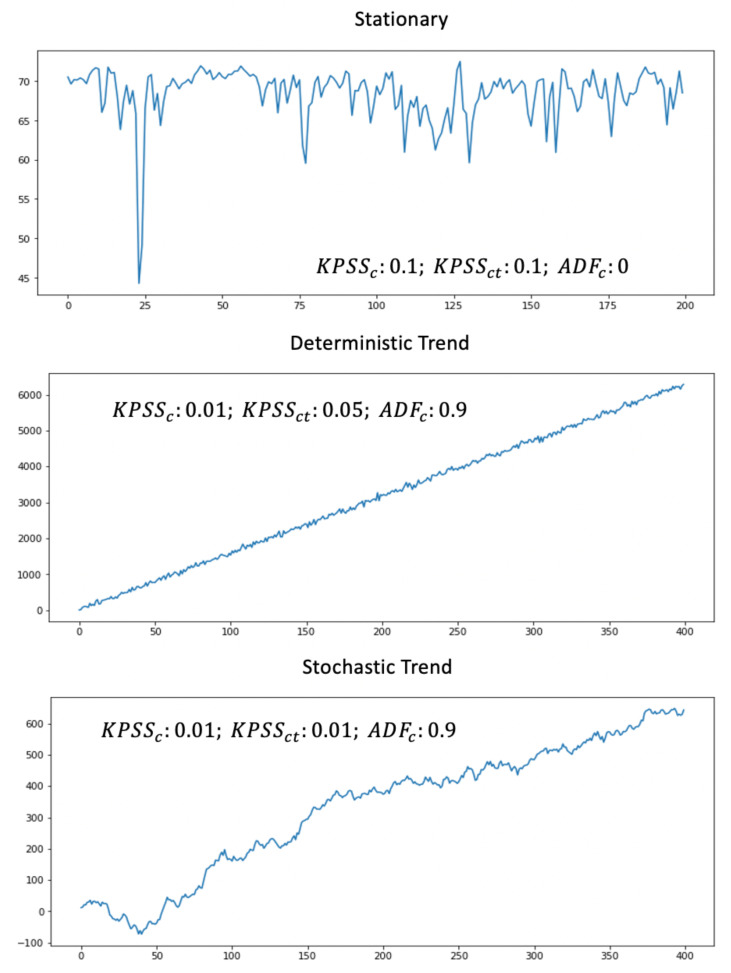
The process of trend testing.

**Figure 10 sensors-21-01590-f010:**
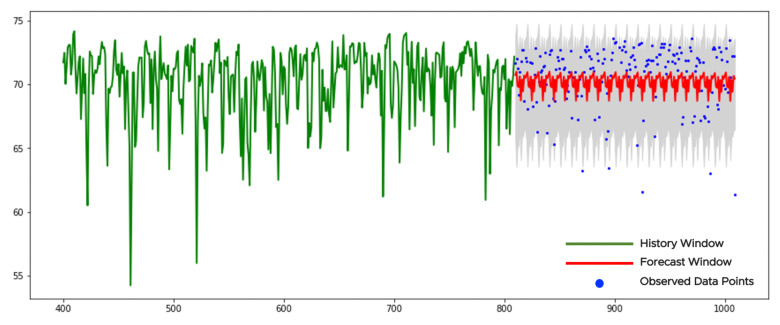
The forecast of a stationary time series.

**Figure 11 sensors-21-01590-f011:**
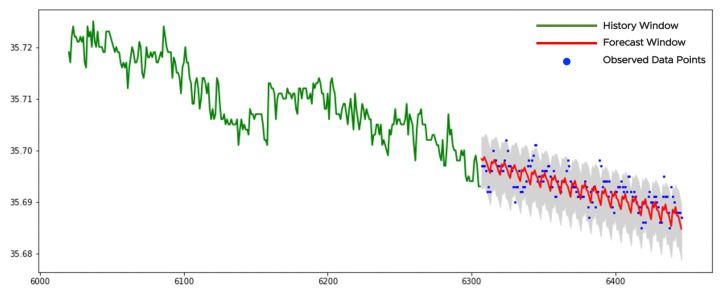
The forecast of a trend-stationary time series.

**Figure 12 sensors-21-01590-f012:**
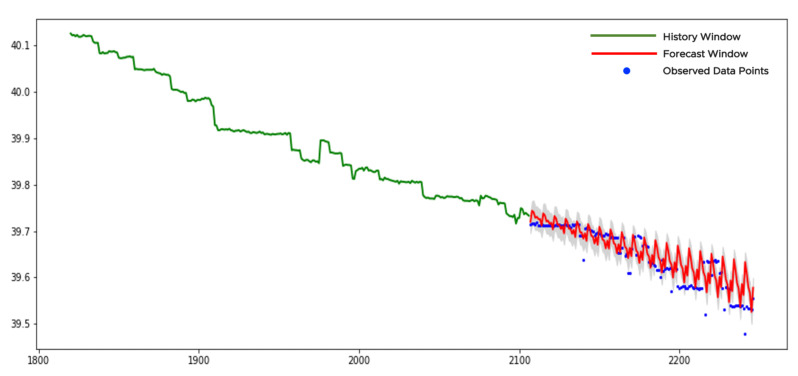
The forecast of a stochastic-trendy time series.

**Figure 13 sensors-21-01590-f013:**
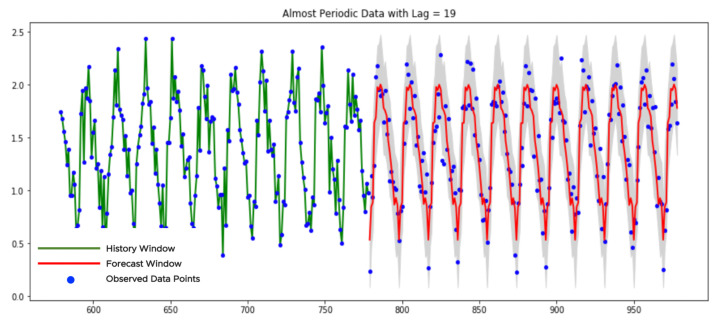
The forecast of a stationary-periodic data.

**Figure 14 sensors-21-01590-f014:**
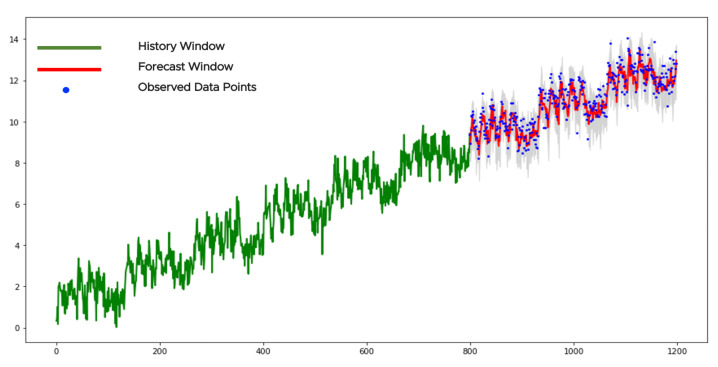
The forecast of a trend-periodic data.

**Figure 15 sensors-21-01590-f015:**
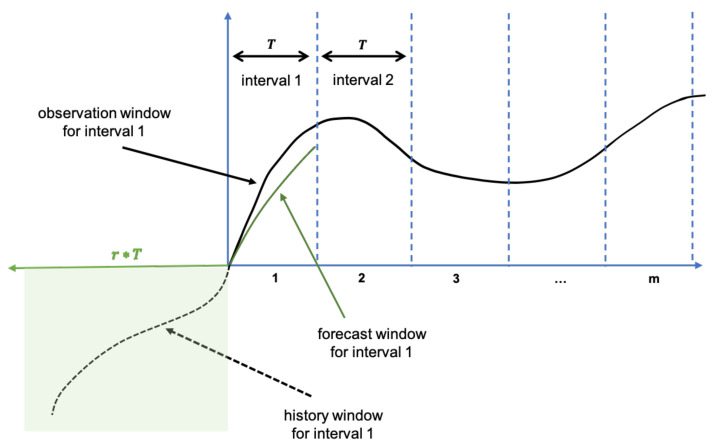
The hidden background of an anomaly monitor.

**Figure 16 sensors-21-01590-f016:**
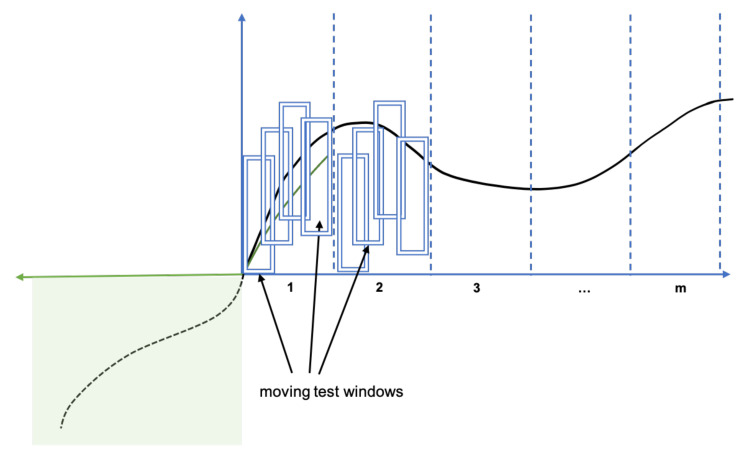
Utilization of test windows for the anomaly signal calculation.

**Figure 17 sensors-21-01590-f017:**
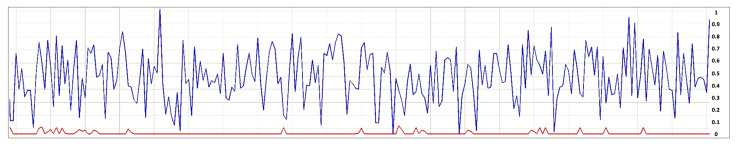
An example of a stationary time series with the anomaly signal.

**Figure 18 sensors-21-01590-f018:**
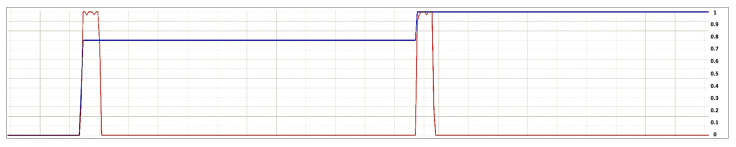
An example of a piecewise-constant time series with the anomaly signal.

**Figure 19 sensors-21-01590-f019:**
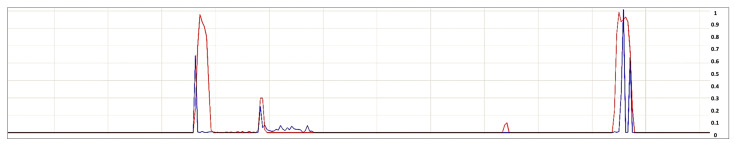
An example of a constant time series with random spikes with the anomaly signal.

**Figure 20 sensors-21-01590-f020:**
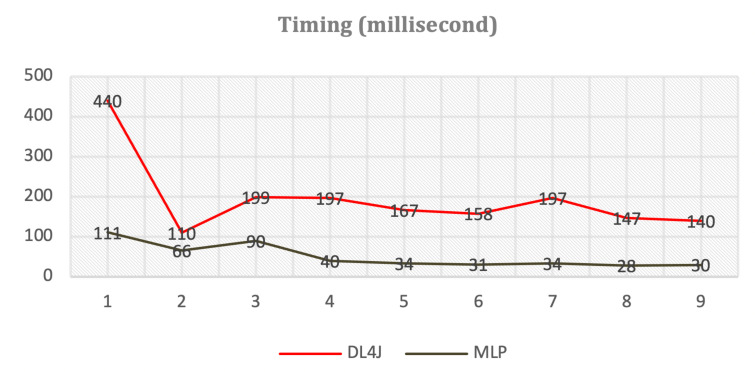
Comparison of different implementations.

**Figure 21 sensors-21-01590-f021:**
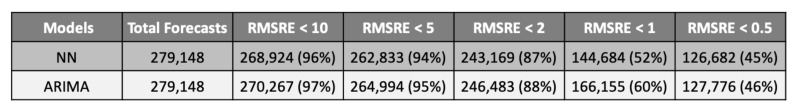
Comparison of root mean square relative errors for different models.

**Figure 22 sensors-21-01590-f022:**
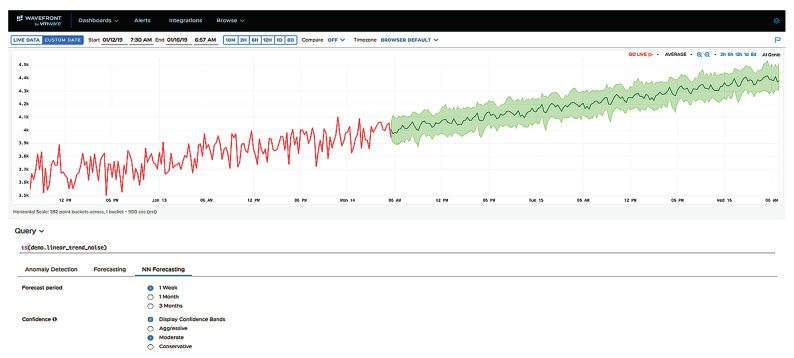
An example of a trend-stationary time series in the Wavefront AI Genie UI.

## Data Availability

The data presented in this study are available on request from the corresponding author. The data are not publicly available due to its size and confidientiality.

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
