# Peer review of "An Enterprise Time Series Forecasting System for Cloud Applications Using Transfer Learning†"

_sensors, 2021, doi:10.3390/s21051590_

Round 1

Reviewer 1 Report

Please see included word document for the authors

Author Response

Thanks for helping to improve the paper. We performed intensive corrections based on your recommendations. More details see in the attached file.

Reviewer 2 Report

What is abbreviation AI/ML in the abstract stands for? i assume Artificial Intelligence/Machine learning ?

-Figure 1 is almost identical to the one in the paper the authors have previously published. 

-Figure 7 have very tiny font. Almost unreadable. The same applied for Figure 9.

- Can you comment more on what the reader should observe in Figure 12? In the text you have put the same text as in the caption of the figure. I think additional details should be given to increase readability. The same applies for the Figures 11 and 14: the tet, as well as the caption of the figure, should be further detailed.

- The font in Figures 17 and 18 is too small. 

Author Response

(The authors gave the same response as above.)
